# The Rise of Parameter Specialization for Knowledge Storage in Large Language Models

**Yihuai Hong**[1,2*]   **Yiran Zhao**[3]   **Wei Tang**[1]   **Yang Deng**[4]   **Yu Rong**[1]   **Wenxuan Zhang**[5†]

[1]Alibaba DAMO Academy  [2]New York University
[3]National University of Singapore
[4]Singapore Management University  [5]Singapore University of Technology and Design

yihuaihong@nyu.edu, wxzhang@sutd.edu.sg

## Abstract

Over time, a growing wave of large language models from various series has been introduced to the community. Researchers are striving to maximize the performance of language models with constrained parameter sizes. However, from a microscopic perspective, there has been limited research on how to better store knowledge in model parameters, particularly within MLPs, to enable more effective utilization of this knowledge by the model. In this work, we analyze twenty publicly available open-source large language models to investigate the relationship between their strong performance and the way knowledge is stored in their corresponding MLP parameters. Our findings reveal that as language models become more advanced and demonstrate stronger knowledge capabilities, their parameters exhibit increased specialization. Specifically, parameters in the MLPs tend to be more focused on encoding similar types of knowledge. We experimentally validate that this specialized distribution of knowledge contributes to improving the efficiency of knowledge utilization in these models. Furthermore, by conducting causal training experiments, we confirm that this specialized knowledge distribution plays a critical role in improving the model's efficiency in leveraging stored knowledge.

## 1  Introduction

An increasing number of powerful large language models (LLMs) have emerged in recent years (Touvron et al., 2023a; Achiam et al., 2023; Groeneveld et al., 2024; Bai et al., 2023; Team, 2025), often demonstrating remarkable capabilities across various benchmarks and tests (Hendrycks et al., 2021a; Chen et al., 2021a; Cobbe et al., 2021). Thanks to the large parameter space, they have shown an exceptional ability to encode vast amounts of knowledge within their parameters, enabling superior performance on knowledge-intensive tasks (Hendrycks et al., 2021a; Zhang et al., 2023).

To understand the internal mechanism of knowledge storage, many studies have been conducted. For example, Geva et al. (2021b) interprets the MLP layers of the transformer architecture (Vaswani et al., 2017) as key-value memories, where the factual knowledge encoded in the weights is retrieved and transmitted to the output layer during inference (Geva et al., 2023; Meng et al., 2022; Yu et al., 2024). Furthermore, researchers have observed that, in the final layer of the MLP, each vector in that value matrix can act as a fundamental unit of knowledge storage (Geva et al., 2022a,b). However, there has been limited research on how to better store and compress knowledge within constrained model parameters to enable more effective utilization of that knowledge by the model.

---

[*]Work done during an internship at Alibaba Group, before joining New York University.
[†]Corresponding author.

39th Conference on Neural Information Processing Systems (NeurIPS 2025).

In this work, we investigate the relationship between language models' knowledge storage patterns and their performance. To identify parameters associated with specific knowledge concepts, we analyze consistently activated parameters in MLP layers when the model processes questions related to the particular knowledge concept. Building on the key-value interpretation of the MLP by Geva et al. (2021b), which treats the up-projection matrix as the key and the down-projection matrix as the value (*i.e.*, stored knowledge), we extract the intermediate representations between these two matrices and treat their absolute value as the activation of corresponding parameters. To support empirical analysis, we construct a new encyclopedic knowledge benchmark based on Wikipedia, covering knowledge concepts with varying frequencies. We then apply the knowledge parameter identification method to 20 open-source LLMs across a wide range of model families, enabling us to explore correlations between knowledge storage patterns and overall model performance.

Our extensive empirical analysis reveals that **stronger models exhibit higher parameter specialization for distinct knowledge**, whereas weaker models distribute knowledge more diffusely across parameters. Consequently, significantly more parameters are required to store individual knowledge in weaker models. As illustrated in Figure 1, advancing model capability correlated with improved parameter specialization for encoding knowledge: fewer parameters are allocated per knowledge concept, while each parameter governs a narrower subset of concepts.

Figure 1: Evolution of knowledge distribution in model parameters during three iterations of LLaMA models. Each parameter vector corresponds to a column in the value matrix of the MLP module, as indicated by the dashed rectangles.

Motivated by this observation, we further conduct four sets of controlled experiments, each involving continued training on the Llama2-7B (Touvron et al., 2023a) and Qwen2-7B (Yang et al., 2024) models with new knowledge respectively, to validate the strong causal relationship between improved parameter specialization and enhanced performance of the models on knowledge tasks. Overall, the experiments reveal that encoding similar knowledge into the same parameter vectors better aligns with the model's internal knowledge retrieval mechanism. This approach helps the model utilize knowledge more efficiently, improves knowledge compression, and reduces hallucination generation.

Our contributions can be summarized as follows:

- To the best of our knowledge, this is the first attempt to quantify and compare the degree of parameter specialization for knowledge storage across different LLMs.

- We investigate the relationship between parameter specialization and model performance in LLMs, constructing a dedicated probing dataset for an in-depth analysis on 20 open-source LLMs. Our findings indicate that more capable LLMs exhibit greater parameter specialization.

- Through controlled training experiments, we provide empirical evidence of a causal link between increased parameter specialization and improved performance on knowledge-intensive tasks.

## 2   Related Work

**Knowledge Storage in LLMs**   Studying how knowledge is stored and utilized in LLMs has been an important area in the research of LLM interpretability (Meng et al., 2022; Geva et al., 2021b; Sukhbaatar et al., 2015; Geva et al., 2023). Recent studies have shown that MLPs are the primary and crucial components for storing factual knowledge and associations in transformer-based language models (Geva et al., 2022b; Dar et al., 2023). They can be conceptualized as key-value memories (Geva et al., 2021b), where the factual knowledge encoded in the MLP weights is recalled and transmitted to the output layer during inference (Geva et al., 2023; Meng et al., 2022; Yu et al., 2024). Additionally, researchers have found that in the final layer of the MLP, each vector in the value matrix can serve as a fundamental unit for storing knowledge (Geva et al., 2022a,b). They have also verified

that by directly manipulating or disrupting these parameter vectors, specific knowledge can be edited or unlearned (Hong et al., 2024a,b; Meng et al., 2022), leading to changes in the model's responses.

**Knowledge Superposition in LLM** Elhage et al. (2022); Olah (2023) propose the concept of Knowledge Superposition. It refers to an inevitable phenomenon in neural network models, especially large language models, during training and data memorization: since the number of data features greatly exceeds the number of parameters in the model, each parameter does not have a simple one-to-one mapping with the data features or knowledge. Neurons are often involved with multiple data features simultaneously. In our work, we treat each vector in the last layer of MLP as a basic unit for storing knowledge and investigate the superposition of knowledge within these vectors.

# 3 Parameter Specialization Analysis for Knowledge Storage

## 3.1 Preliminary

In transformer-based language models, the MLP is a crucial component for storing the model's factual knowledge, and its sub-layers can be viewed as key-value memories (Geva et al., 2021b). To be specific, the first layer* of MLP sublayers can be viewed as a matrix $W_K$ formed by key vectors $\{\mathbf{k}_1, \mathbf{k}_2, \ldots, \mathbf{k}_n\}$, used to capture a set of patterns in the input sequence, and ultimately outputting the coefficient scores. The second layer can be viewed as a matrix $W_V$ formed by value vectors $\{\mathbf{v}_1, \mathbf{v}_2, \ldots, \mathbf{v}_n\}$, with each value vector containing the corresponding factual knowledge.

Formally, the output of the MLP in the transformer's $\ell$-th layer, given an input hidden state $\mathbf{x}^\ell$, can be defined as:

$$\mathbf{M}^\ell = f\big(W_K^\ell \cdot \gamma(\mathbf{x}^\ell + \mathbf{A}^\ell)\big)W_V^\ell = \mathbf{m}^\ell W_V^\ell, \tag{1}$$

where $W_K^\ell, W_V^\ell \in \mathbb{R}^{n \times d}$. The function $f$ and $\gamma$ represent a non-linearity† and layer normalization, respectively. In the transformer's $\ell$-th layer, $\mathbf{m}^\ell \in \mathbb{R}^n$ denotes the coefficient scores, and $\mathbf{A}^\ell$ represents the output of the attention component. The hidden state dimension is $d$, while the intermediate MLP has a dimension of $n$. Then, by denoting $\mathbf{v}_j^\ell$ as the $j$-th column (which will be called the value vector or parameter vector in the following sections) of $W_V^\ell$ and $m_j^\ell$ as the $j$-th element in the coefficients produced by the first layer of the MLP, we can view MLP's output $\mathbf{M}^\ell$ as a linear combination of the value vectors in $W_V^\ell$, with their corresponding coefficients $\mathbf{m}^\ell$:

$$\mathbf{M}^\ell = \sum\nolimits_{j=1}^n m_j^\ell \mathbf{v}_j^\ell, \tag{2}$$

Finally, the hidden states at the $\ell$-th layer of the language model can be defined as:

$$X^{\ell+1} = X^\ell + \mathbf{M}^\ell + \mathbf{A}^\ell, \tag{3}$$

where $X^\ell$, $\mathbf{M}^\ell$ and $\mathbf{A}^\ell$ represent the hidden states, MLP's output, and the attention component's output in the transformer's $\ell$-th layer, respectively. In this work, we focus on studying the impact of the MLP on the knowledge output of the hidden states.

## 3.2 Knowledge Vectors Masking Procedure

Referring to Eq. (2), if we aim to ablate the impact of the knowledge contained in the vectors for a particular subset $S^\ell$ of indices in $\ell$-th layer, we can directly set the corresponding $m_j^\ell$ values for $j \in S^\ell$ to zero. Hence, we have:

$$\mathbf{M}_{\text{masked}}^\ell = \sum\nolimits_{\substack{j=1 \\ j \notin S^\ell}}^n m_j^\ell \mathbf{v}_j^\ell + \sum\nolimits_{\substack{j=1 \\ j \in S^\ell}}^n 0 \cdot \mathbf{v}_j^\ell = \sum\nolimits_{\substack{j=1 \\ j \notin S^\ell}}^n m_j^\ell \mathbf{v}_j^\ell, \tag{4}$$

---

*In most decoder-only models, such as GPT-2 (Radford et al., 2019) and GPT-J (Chen et al., 2021b), the MLP component consists of two layers, whereas in LLaMA (Touvron et al., 2023b), it comprises three layers. However, we can still regard LLaMA's first two layers collectively as the key matrices, with their output representing the coefficient scores.

†For brevity, the bias term is omitted.

Therefore, given a concept, when we aim to identify which specific value vectors in the model's MLPs are most closely related to the knowledge contained in that concept—while avoiding the masking of vectors associated with the model's general capabilities[‡] (Meng et al., 2022; Geva et al., 2023), i.e., determining the appropriate subset $S^\ell$ at each layer of the model for this concept—we will run $t$ concept-related questions and $t^*$ irrelevant questions on the selected model. Then we will compute the corresponding coefficients $\mathbf{m}^\ell$ and $\mathbf{m}^{*\ell}$, which are the averages of the coefficients for the concept-related questions and irrelevant questions, respectively, at each layer of the model. For details on the generation of concept-related and irrelevant questions, as well as the selection of $t$ and $t^*$, please refer to §3.4. After obtaining $\mathbf{m}^\ell$ and $\mathbf{m}^{*\ell}$ at each layer, we perform the computation using the following formula:

$$\mathbf{S}^\ell = \left\{ \left| m_j^\ell - m_j^{*\ell} \right| \,\middle|\, 1 \leq j \leq n, \; m_j^\ell \in \mathbf{m}^\ell, \; m_j^{*\ell} \in \mathbf{m}^{*\ell} \right\} \tag{5}$$

Next, we will sort $\mathbf{S}^\ell$ in descending order and select the value vectors corresponding to the indices of the top $k$ elements, which will be used as the subset $S^\ell$ for the masking operation. This allows us to observe and analyze the impact of masking these vectors on the model's knowledge output for certain concepts.

## 3.3 The Definition of Parameter Specialization

After obtaining the subset of value vectors $S^\ell$ that exhibit specificity to a given concept at each layer of the model, as described in §3.2, we apply the masking operation to these value vectors, as shown in Eq. (4). We then analyze its impact on the model's final outputs for the $t$ concept-related questions and $t^*$ irrelevant questions. By comparing the model's responses after masking with the ground truth answers, we compute the accuracy on concept-related questions, referred to as the Concept Specific Score after surgery, and the accuracy on irrelevant questions, referred to as the General Score after surgery. To quantify the degrees of specialization of the model's value vectors with respect to the concept-related knowledge, we define the **Parameter Specialization Score (PSS)**:

$$\text{PSS} \triangleq \frac{\left| \text{General Score after surgery} - \text{Concept Specific Score after surgery} \right|}{\text{General Score before surgery}}, \tag{6}$$

which is obtained by taking the absolute difference between the General Score and the Concept Specific Score after surgery, and then dividing by the model's accuracy on the entire dataset before surgery. A higher PSS indicates that the parameter vectors in the model's MLP layers exhibit a higher degree of specialization towards specific knowledge. Conversely, a lower PSS suggests more severe knowledge superposition phenomena within the parameter vectors, resulting in a lower degree of specialization.

## 3.4 Dataset Construction

To thoroughly investigate the parameter specialization of knowledge with different frequencies in the parameter vectors of LLMs' MLP, we introduce a dataset named SpecWiki. It includes 525 concepts selected from Wikipedia [§], a widely recognized high-quality corpus for LLM training. These concepts are categorized based on their frequency levels to ensure a diverse distribution. We then design two distinct question formats—multiple-choice questions and open-ended generation prompts—to facilitate a thorough examination of the models' knowledge storage.

**Concept Selection** We treat each Wikipedia item as a defining concept, typically represented by an article focused on a specific subject, indicated by its title. We focus on specific entity concepts, such as historical figures, events and locations. We began by randomly sampling 2,400 pages (a 0.01% rate) from the 2019 version of Wikipedia. Subsequently, we performed manual filtering to remove overly commonsensical or abstract concepts (such as the letter 'S' and the word 'Freedom'), ambiguous concepts (like 'Apple'), and those associated with pages under 1,000 words. Ultimately, this resulted in 525 high-quality concepts spanning specific topics like people, arts, and events.

---

[‡]The term "general ability" refers to the model's fundamental skills, such as processing text inputs correctly and generating coherent outputs, rather than encoding knowledge specific to a particular concept.

[§]https://en.wikipedia.org/

Given that the frequency of knowledge in training datasets significantly influences a model's ability to retain and comprehend it (Allen-Zhu & Li, 2023; Meng et al., 2022; Mallen et al., 2023), we utilize Wikipedia page views as a proxy for knowledge frequency in the models' pre-training datasets[¶]. To this end, we calculated the page views for each concept on Wikipedia between January 1, 2010, and December 31, 2019[‖]. Based on these statistics, concepts are categorized by page view frequency into three equal tiers: low-frequency (bottom 33% of the distribution), medium-frequency (middle 33%), and high-frequency (top 33%). A more detailed distribution of the categories and the corresponding example data of SpecWiki dataset are provided in Table 4 and Table 5, respectively, in the Appendix. This approximation helps estimate the likelihood of a concept's presence in the models' pre-training datasets and allows us to explore how knowledge at different frequency levels is stored in models and provides a more comprehensive evaluation.

**Question Generation** To more precisely assess the retention of knowledge within the model, we design two sets of question formats.

- *Multi-Choice Questions*. Drawing inspiration from the widely used Massive Multitask Language Understanding (MMLU) dataset (Hendrycks et al., 2021b), which evaluates general knowledge across models, we similarly designed ten multiple-choice questions for each concept, ensuring the knowledge and answers could be directly found in the relevant Wikipedia articles. Specifically, we provided GPT-4o (OpenAI et al., 2024) with the appropriate Wikipedia article for each concept and instructed it to extract ten questions without overlap, along with the correct answers derived from the article's text. Next, it was instructed to generate three additional incorrect answers, aside from the golden answer, ensuring that none of them overlapped with the correct answer to avoid confusion. The detailed prompt is available in §A.1. We also include sample multiple-choice questions and results of manual verification of the generated data in Appendix §A.2.

- *Open-ended Generation*. To more effectively assess the model's ability to generate knowledge text freely, and to overcome the randomness and lack of depth inherent in the Multi-Choice Question evaluation method, we also set up a series of Open-ended Generation questions. For each question related to a concept, we prompted the model to generate an answer of up to 150 tokens directly and used GPT-4o as an evaluator to evaluate whether the generated response correctly matched the golden answer.

## 4 Experiment

### 4.1 Experimental Setup

**Evaluated Models** To provide a more comprehensive evaluation of how the degree of parameter specialization evolves across large language models, we assessed 20 open-source models from various families and sizes in the community. Specifically, we evaluated LLaMA series (Touvron et al., 2023a,b; Grattafiori et al., 2024), Qwen series (Bai et al., 2023; Yang et al., 2024; Team, 2025), Gemma series (Team et al., 2024a,b), OLMo series (Groeneveld et al., 2024; OLMo et al., 2025), Yi series (AI et al., 2025), Mistral series (Jiang et al., 2023), GPT-j-6b (Wang & Komatsuzaki, 2021), Pythia-6.9b (Biderman et al., 2023), Falcon-7b (Almazrouei et al., 2023) and Mpt-7b (Databricks, 2023). Refer to Appendix §B.1 for the implementation details of these models.

**Knowledge Vectors Masking Setup** Based on the descriptions in §3.2, in order to obtain $\mathbf{m}^{\ell}$ and $\mathbf{m}^{*\ell}$ for each concept in SpecWiki at each layer of the model, we set the number of concept-related questions $t$ to 10. Additionally, we randomly select 5 irrelevant concepts with no knowledge overlap from the benchmark, and gather the corresponding questions associated with these irrelevant concepts, resulting in $t^* = 50$ irrelevant questions. These collected questions will also be directly utilized in the computation of both the Concept-Specific Score and the General Score.

---

[¶]To better support this point, we include experiments in §A.4 of Appendix that validate the strong correlation between concept popularity and their frequency in the pretraining data.

[‖]The earliest release date of the evaluated models, such as GPT-J, is 2019. Therefore, their pretraining datasets could not include knowledge or concepts that emerged after this period. To ensure a fair evaluation across all models, any knowledge introduced post-2019, including the COVID-19 pandemic, was excluded from the benchmark.

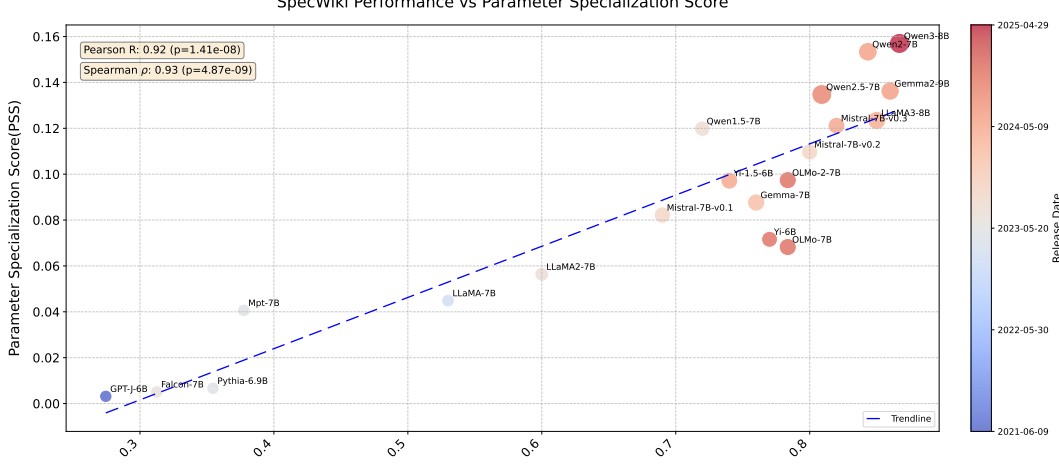

Figure 2: Correlation between the performance on SpecWiki and parameter specialization score (PSS) in 20 language models. We use a color gradient to distinguish the release times of the models, with cooler colors indicating earlier release dates and warmer colors representing later releases. Additionally, the size of each circle reflects the model's performance on MMLU, with larger circles indicating better performance. The blue trendline, obtained through linear regression fitting of the data points, suggests a strong correlation between a model's performance on SpecWiki and its degree of Parameter Specialization.

Regarding the selection of model layers for masking, since the initial layers of a model typically handle fundamental capabilities like basic text processing (Meng et al., 2022; Geva et al., 2023), masking these layers could severely impair the model's basic text generation abilities. Therefore, for all models in our study, we preserve the first 5 layers without masking and only apply vector masking operations to all subsequent layers.

For the PSS computation of each model, we selected five different fixed $k$ values—10%, 20%, 30%, 40%, and 50%—which represent the proportion of value vectors in the model's MLP layers that were masked. For each k, we calculated the corresponding PSS following 6, and then averaged the results to obtain the final PSS score for each model. This criterion was applied consistently across both the Multiple-Choice Questions (MCQ) and Open-ended Generation (OEG) tasks.

## 4.2 Main Results

The main results for the Multiple-Choice Questions setting can be seen in Figure 2. We observe a strong correlation between the degree of Parameter Specialization (measured by PSS) and model performance on SpecWiki across 20 models, with Pearson and Spearman coefficients of 0.92 and 0.93, respectively. Models achieving better performance on SpecWiki exhibit higher Parameter Specialization Scores. Furthermore, models with higher PSS are often those released more recently (warmer color) and exhibit stronger general abilities, as measured by their MMLU performance (larger circle). The corresponding results for the Open-ended Generation setting can be found in Figure 6 in §B.2, which exhibit similar patterns and trends.

To better analyze the variations in Parameter Specialization across models within the same family, we selected eight models from four model families: LLaMA, Qwen, Mistral, and Gemma. We examined how the difference between the General Score, which represents the model's ability to handle irrelevant knowledge, and the Concept Specific Score, which reflects the model's ability to handle task-specific knowledge, changes under different masking ratios of parameter vectors. The results are shown in Figure 3.

From the figure, we can observe a very similar pattern across models from the four families:

1. Among models within the same family, more advanced models tend to achieve higher peaks in the General Score - Concept Specific Score difference. This indicates that more advanced models generally exhibit higher levels of Parameter Specialization.

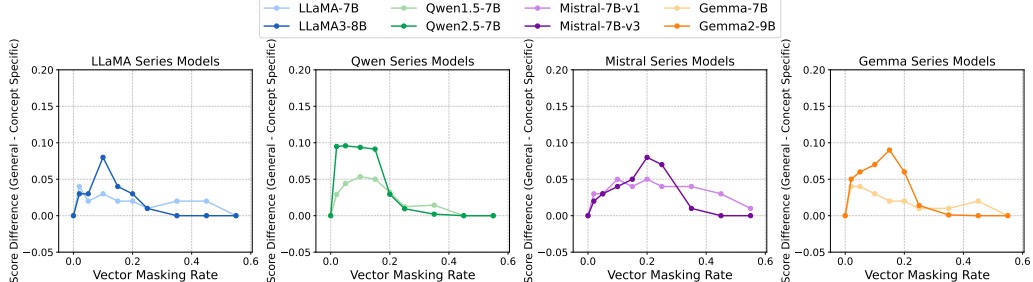

Figure 3: Analysis of Parameter Specialization variations across models within the same family. We selected eight models from four model families: LLaMA, Qwen, Mistral, and Gemma. The figure shows how the difference between the General Score (representing the model's ability to handle irrelevant knowledge) and the Concept Specific Score (representing the model's ability to handle task-specific knowledge) changes under different masking ratios of parameter vectors.

2. As the masking ratio of parameter vectors increases, from approximately 5% to 20%, the difference between the General Score and the Concept Specific Score gradually increases to a peak. This indicates that we are removing parameter vectors that are highly specific to the target knowledge. After reaching the peak, as the masking ratio continues to increase, the difference gradually decreases to zero. This suggests that parameter vectors with lower activation are often those that have a higher degree of knowledge superposition and are less specialized in the target knowledge.

Additionally, we unexpectedly found that when a small proportion of concept-related vectors (ranging from 5% to 10%) were masked, the performance of the masked models on unrelated questions even surpassed that of the original models. This observation is consistent across various models and indicates the positive impact of reducing irrelevant information interference in the model's representation, leading to improved performance.

In §5, we will further validate the causal relationship between the degree of model parameter specialization and its ability to better utilize target knowledge through the finetuning experiments on additional data.

### 4.3 Impact of Model Scale on Parameter Specialization

In this section, to better explore the differences in the degree of parameter specialization across models of different sizes, we conducted Knowledge Vectors Masking experiments on five Qwen1.5 models of varying sizes (0.5B, 1.8B, 4B, 7B, and 14B) and on 2 Gemma2 models (2B and 9B). The results are shown in Table 1. We observe that in both the Qwen and Gemma model families, as the model size increases, the corresponding Parameter Specialization Score also increases. This trend is accompanied by improved performance on SpecWiki. This suggests that in larger-scale models, the degree of superposition for specific knowledge decreases and it tends to be distinctly represented across designated parameter vectors.

| Model | Accuracy$_{\text{MCQ}}$ ↑ | PSS ↑ |
|---|---|---|
| Qwen1.5-0.5B | 0.61 ($\pm$0.2) | 0.019 ($\pm$0.01) |
| Qwen1.5-1.8B | 0.61 ($\pm$0.3) | 0.044 ($\pm$0.02) |
| Qwen1.5-4B | 0.73 ($\pm$0.2) | 0.075 ($\pm$0.02) |
| Qwen1.5-7B | 0.75 ($\pm$0.2) | 0.121 ($\pm$0.04) |
| Qwen1.5-14B | **0.82** ($\pm$0.2) | **0.184** ($\pm$0.03) |
| Gemma2-2B | 0.72 ($\pm$0.3) | 0.057 ($\pm$0.02) |
| Gemma2-9B | **0.86** ($\pm$0.1) | 0.138 ($\pm$0.03) |

Table 1: Performance comparison of language models with varying sizes on Multiple-Choice Question and Parameter Specialization Score. Both the Qwen1.5 and Gemma2 series models show improved Parameter Specialization as the model scale increases, accompanied by better performance on the MCQ testing in SpceWiki.

### 4.4 Evolution of Parameter Specialization During Pretraining

To better investigate the development of Parameter Specialization in the model from the perspective of model training dynamics, we analyzed 10 checkpoints from the OLMo-2-1124-7B (OLMo et al., 2025) pretraining process by using our SpecWiki. The results are shown in Figure 4 below.

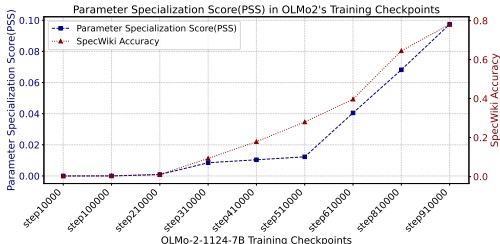

Figure 4: Development of Parameter Specialization in OLMo-2-1124-7B over the pretraining process.

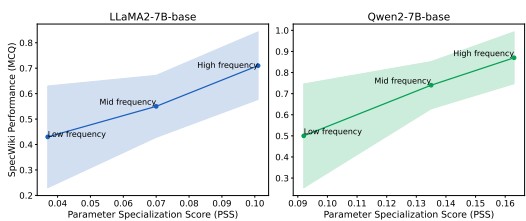

Figure 5: Relationship between concept popularity, model accuracy on MCQ, and Parameter Specialization Score in LLaMA2-7B and Qwen2-7B models.

| Model | Accuracy$_{MCQ}$ ↑ | Accuracy$_{OEG}$ ↑ | PSS ↑ | Semantic Entropy ↓ | Local Intrinsic Dimension ↓ |
|---|---|---|---|---|---|
| LLaMA2-7B | 0.60 (±0.2) | 0.51 (±0.1) | 0.67 (±0.1) | 0.67 (±0.1) | 11.23 (±2.1) |
| LLaMA2-7B$_{FT-FV}$ | 0.63 (±0.3) | 0.54 (±0.2) | 0.65 (±0.2) | 0.62 (±0.1) | 11.12 (±1.4) |
| LLaMA2-7B$_{FT-PV}$ | **0.67** (±0.3) | **0.59** (±0.2) | **0.72** (±0.1) | **0.50** (±0.2) | **7.89** (±2.1) |
| LLaMA2-7B$_{FT-CV}$ | 0.62 (±0.1) | 0.51 (±0.1) | 0.63 (±0.2) | 0.62 (±0.1) | 11.12 (±1.4) |
| LLaMA2-7B$_{FT-RV}$ | 0.58 (±0.2) | 0.49 (±0.2) | 0.65 (±0.2) | 0.65 (±0.2) | 11.07 (±2.7) |
| Qwen2-7B | 0.72 (±0.3) | 0.63 (±0.1) | 0.124 (±0.03) | 0.56 (±0.1) | 9.78 (±1.9) |
| Qwen2-7B$_{FT-FV}$ | 0.73 (±0.2) | 0.67 (±0.2) | 0.110 (±0.02) | 0.59 (±0.2) | 8.53 (±1.1) |
| Qwen2-7B$_{FT-PV}$ | **0.77** (±0.1) | **0.70** (±0.1) | **0.133** (±0.03) | **0.39** (±0.1) | **6.92** (±1.3) |
| Qwen2-7B$_{FT-CV}$ | 0.73 (±0.2) | 0.65 (±0.2) | 0.114 (±0.03) | 0.55 (±0.2) | 8.78 (±1.6) |
| Qwen2-7B$_{FT-RV}$ | 0.71 (±0.2) | 0.63 (±0.1) | 0.122 (±0.02) | 0.59 (±0.2) | 9.65 (±1.4) |

Table 2: The performance of both the original LLaMA2-7B-base and Qwen2-7B-base models, along with their FT-FV, FT-PV, FT-CV and FT-RV variants, was assessed on a selection of 10 high-frequency concepts from SpecWiki. Five metrics were used to evaluate their performance, including their general effectiveness, Parameter Specialization, and the degree of hallucination present in their output.

From the results, we observe that during the early training steps (step 10,000 to step 210,000), both the PSS and the accuracy on SpeciWiki remain nearly unchanged and close to zero. In the subsequent phase (step 310,000 to step 510,000), although the model begins to show noticeable gains in accuracy on SpeciWiki, the PSS are still under 0.1. However, it is during the later training steps (step 610,000 to step 910,000) that parameter specialization begins to emerge, accompanied by a more substantial improvement in accuracy. These findings suggest that parameter specialization does not occur in the early stages of training, but rather emerges after a certain amount of data exposure. Furthermore, as training continues and the model sees more data, the degree of parameter specialization increases accordingly.

### 4.5 Parameter Specialization in Relation to Concept Popularity

In this section, we analyze how the popularity of concepts themselves, which is roughly equivalent to their frequency in the pretraining data, will affect the level of parameter specialization for the corresponding knowledge. We follow the classification method for concepts as outlined in §3.4, dividing them into high-frequency, mid-frequency, and low-frequency categories. The impact on their PSS scores is measured on two example models, LLaMA2-7B, and Qwen2.5-7B, which are shown in Figure 5.

From the figure, it is clear that in both the LLaMA2-7B and Qwen2-7B models, as the popularity of a concept decreases, the model's accuracy on that specific knowledge declines, accompanied by a lower Parameter Specialization Score. This suggests that the degree of Parameter Specialization for a particular knowledge in the model's parameters is likely directly correlated with the frequency of that knowledge in the model's pretraining dataset. The higher the frequency, the greater the Parameter Specialization for that knowledge in the model.

## 5 Validation of Parameter Specialization Benefits for Knowledge Tasks

In this section, we conducted four sets of controlled training experiments, each involving continued fine-tuning on the Llama2-7B-base (Touvron et al., 2023a) and Qwen2-7B-base (Yang et al., 2024)

models with additional knowledge data. These experiments aim to validate the causal relationship between improved parameter specialization and enhanced model performance on knowledge tasks.

## 5.1 Finetuning Setup

We randomly selected 10 high-frequency concepts from the SpecWiki benchmark. For each concept, we gathered relevant textual material from the top 10 most popular Google search results, including the corresponding Wikipedia article, and compiled this into an additional finetuning training dataset.

Next, we will validate whether the improvement in Parameter Specialization and the enhanced efficiency in the model's use of knowledge truly exhibit a causal relationship through four distinct finetuning experiments. The experimental setups are detailed below:

**FT-FV(Full Vectors)** Perform full finetuning (FT) on all parameter vectors of the MLPs across all layers in the model, while keeping the other parameters frozen.

**FT-PV(Partial Vectors)** Perform partial finetuning on a subset of the parameter vectors in the MLPs while keeping the other parameters frozen. Specifically, for each model, we apply finetuning (FT) to the top $\frac{k}{8}$ most highly activated parameter vectors[**]. For the selection of $k$, please refer to the description in §4.1.

**FT-CV(Complementary Vectors)** Perform finetuning only on the complementary set of parameter vectors, excluding the target vectors.

**FT-RV(Random Vectors)** Perform finetuning on a subset of parameter vectors randomly selected from the MLP, ensuring the same quantity as in the FT-PV setting.

## 5.2 Finetuning Results

In addition to evaluating the model's performance on SpecWiki's Multi-choice Question and Open-ended Generation tests, as well as the Parameter Specialization scores, we also report two other metrics, Semantic Entropy (Kuhn et al., 2022) and Local Intrinsic Dimension (LID) (Yin et al., 2024), for measuring the extent of hallucination in the model's output. These metrics help evaluate whether training strategies that enhance Parameter Specialization—by aligning better with the model's knowledge retrieval mechanisms through a data-encoded strategy—can effectively reduce the unintended side effect of hallucination. For a detailed introduction to these two hallucination measurements, please refer to §B.3.

The final results are presented in Table 2. From the results, we can see that the FT-PV method, which finetunes only a small subset of the highly activated knowledge parameters, not only further enhances the model's Parameter Specialization compared to the three other finetuning setups, but also greatly improves the model's utilization of specific knowledge. As a result, it achieves the best performance on the benchmark Multi-Choice questions and Open-ended generation tasks. Additionally, by reducing the influence of irrelevant information in the model's key parameter vectors, FT-PV helps to significantly reduce the level of hallucination in the generated text.

Although FT-FV and FT-CV does improve the model's performance on both the Multi-Choice questions and Open-ended generation tasks to some extent, compared to FT-PV, it does not lead to a better increase in Parameter Specialization. Additionally, the degree of hallucination in the generated text is not effectively reduced. FT-RV, serving as a counterpart to FT-PV, demonstrates that fine-tuning the same number of arbitrary value vectors in the model's MLP can not result in a desirable knowledge enhancement.

## 6 Conclusion

This study reveals that enhanced parameter specialization—where related knowledge is encoded in focused parameter vectors—correlates with superior performance in large language models. Analyzing 20 open-source models, we observed stronger models increasingly consolidate similar knowledge into fewer parameters, while weaker models distribute it diffusely. Controlled experiments confirmed

---

[**]We experimented with $\frac{k}{2}$, $\frac{k}{4}$, $\frac{k}{8}$, and $\frac{k}{16}$, and found that finetuning only $\frac{k}{8}$ of the parameter vectors was sufficient to achieve excellent performance.

that optimizing this specialization improves task performance and reduces hallucination. These findings highlight the importance of aligning knowledge storage with models' retrieval mechanisms for efficiency and accuracy. Future work should explore dynamic knowledge updates and scalability, advancing both interpretability and performance in LLM design.

## 7 Limitations and Future Work

In our work, we have only examined and validated knowledge parameter specialization within the MLP, and this was done by treating vectors in the MLP as units of analysis. However, at least this remains one of the knowledge storage methods that has been extensively validated so far (Geva et al., 2021a; Meng et al., 2022; Geva et al., 2023). In fact, knowledge may also reside within the attention module of transformer models (Geva et al., 2023).

Additionally, due to GPU limitations, all the models we tested are smaller than or equal to 14B parameters, so we were unable to validate our conclusions on larger models, such as those with 35B parameters or more.

In future work, we will progressively narrow the focus of our research to individual neurons in language models, aiming to measure and validate more precise Parameter Specialization and Parameter Superposition. In addition, we will extend this concept to the study of other related model architectures, including Mixture of Experts (Fedus et al., 2022), which similarly enhances model performance by specializing expert parameters, as well as Sparse Auto-Encoders (Huben et al., 2024), which help clarify the model's representations by leveraging a larger parameter space and mitigating the superposition of these features.

## Acknowledgment

This research is supported by the Ministry of Education, Singapore, under its Academic Research Fund (AcRF) Tier 1 grant, and funded through the SUTD Assistant Professorship Scheme (SAP 2025_001).

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

# A    Details of Dataset

## A.1    Dataset Construction Prompts

Below is our prompt for querying GPT-4o to generate the options for Multi-Choice questions of each concept:

```
Please provide four answer options (A, B, C, D) for the following
question, and indicate the correct answer. Example: Question: 'When
was Costa Coffee founded?' Options: A) 1971 B) 1985 C) 1992 D) 2000
Correct Answer: A) 1971
Now, please answer the following question: Question: Question
Options:
```

Below is our prompt for querying the model to generate the answers for Multi-choice questions in three-shot setting:

```
**Question:** What is the capital city of France? **Options:** A.
Berlin B. Madrid C. Paris D. Rome **Answer:** C
**Question:** What is the largest planet in our solar system?
**Options:** A. Earth B. Jupiter C. Mars D. Venus **Answer:** B
**Question:** Which element has the chemical symbol "O"? **Options:**
A. Oxygen B. Gold C. Silver D. Iron **Answer:** A
**Question:** question **Options:** A. option a B. option b C. option
c D. option d
```

Below is our prompt for collecting the coefficients in model when querying about concept-reated knowledge:

```
Question: question Answer: answer:
```

## A.2    Manual Verification

Here, we describe the manual verification process used in constructing SpecWiki, including the validation of model-generated data:

Specifically, we analyze a subset of 524 (10%) questions from SpecWiki, by sampling 50% of the concepts and randomly selecting 2 questions per concept. Then, we manually verify that the questions are about the given concept and that they are simple and reasonable. In addition, we review all the generated questions for 200 sampled concepts and verify they are not repetitive. We find that all analyzed questions were about the given concept and that 522 (99%) of them are reasonable simple questions. Moreover, we observe that questions are generally diverse, with only 1 out of 20 concepts having 2 (out of 10) similar questions. This shows that our data generation process produces valid and diverse instances for evaluation.

## A.3    Dataset Categories and Examples

Here, we provided a more detailed distribution of the categories and the corresponding example data of SpecWiki dataset in Table 4 and Table 5, respectively.

## A.4    Validation of Popularity-Frequency Correlation

We included a simple experiment to validate the strong correlation between the popularity of concepts and their frequency in the pretraining data. To be specific, we used The Pile(Gao et al., 2020), which currently serves as a significant portion of the pretraining dataset for most large language models(Touvron et al., 2023a; Groeneveld et al., 2024; Team et al., 2024a), as an example of a pretraining corpus. We then counted the frequency with which each of the 525 concepts from SpecWiki appeared in all text segments of The Pile dataset via the Elasticsearch API(Elazar et al., 2024). Subsequently, we compared these frequencies with the popularity metrics for each concept and

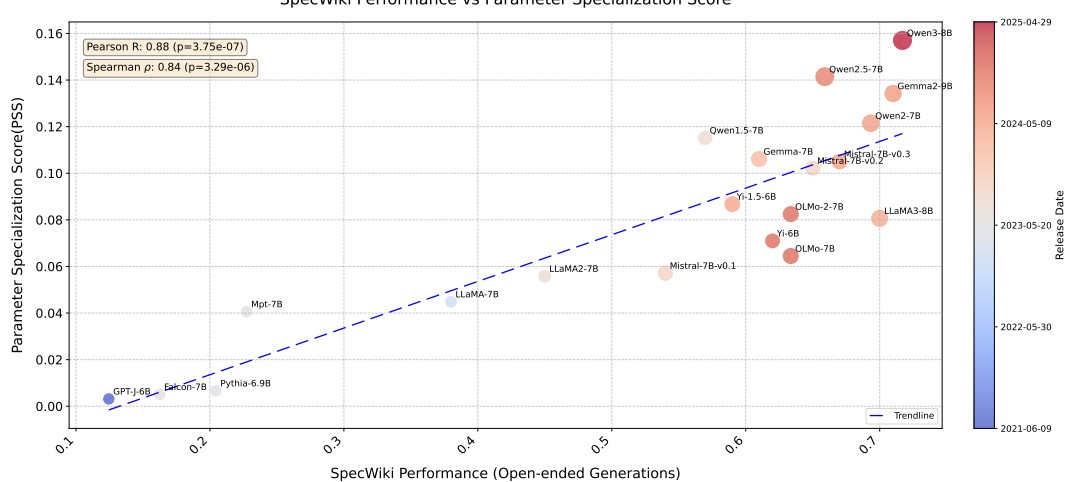

Figure 6: Performance across 20 models on Parameter Specialization Scores on Open-ended Generation Setting.

computed the corresponding Spearman's rank correlation coefficient. The result is 0.814, indicating a strong correlation.

Additionally, Table 3 presents the top 3 most popular and the bottom 3 least popular concept examples in SpecWiki, along with their occurrence counts in The Pile dataset and their corresponding popularity scores.

| Top 3 High Popularity Example Concept | Frequency | Popularity |
|---|---|---|
| Wikipedia | 911708 | 1414686 |
| Barack Obama | 984586 | 1128538 |
| India | 3839241 | 1024513 |
| **Bottom 3 Low Popularity Example Concept** | **Frequency** | **Popularity** |
| Dark Souls (video game) | 49 | 27 |
| Culture of Latin America | 251 | 90 |
| Array (data structure) | 1164 | 94 |

Table 3: Top and bottom 3 concepts ranked by popularity and their corresponding frequencies.

# B  Details of Experiments

## B.1  The implementation of the models

For all models, the inference is performed in a text completion/generation mode, without the addition of any instruction tokens, to better assess the knowledge present in the model. For the Multi-Choice Questions task, we use a three-shot setup for each model and search for the answer within the next 30 tokens generated by the model. For the open-ended generation task, we prompt the model in a zero-shot setting to produce an answer no longer than 150 tokens.

All the experiments in this work were conducted on four 80GB NVIDIA A800 GPUs.

## B.2  Parameter Specialization Scores on OEG Setting

In Figure 6 we provided the performance across 20 models on Parameter Specialization Scores on Open-ended Generation Setting.

| High Frequency (Number of Concepts: 181) | | | | Medium Frequency (Number of Concepts: 191) | | | | Low Frequency (Number of Concepts: 153) | | | |
|---|---|---|---|---|---|---|---|---|---|---|---|
| Country | 13.3% | Technology | 7.6% | Technology | 19.9% | Mathematics | 4.4% | Person | 21.9% | Brand/Product | 6.3% |
| Culture | 9.5% | Brand/Product | 7.6% | Art and Entertainment | 11.1% | Politics | 4.4% | History | 10.6% | Medical | 5.5% |
| Location | 8.6% | Person | 6.7% | Natural Sciences | 10.5% | Location | 4.4% | Entertainment | 8.6% | Culture | 2.9% |
| History | 8.6% | Medical | 6.7% | Medical/Biology | 7.7% | Country | 3.9% | Company/Organization | 7.3% | Others | 2.3% |
| Sports | 7.6% | Entertainment | 6.7% | Culture | 7.2% | Company/Organization | 3.3% | Others | 6.3% | Natural Sciences | 2.1% |

Table 4: Ten most frequent concept categories of SpecWiki in high frequency, medium frequency, and low frequency levels.

| Example Concept | Frequency Level | Category | Example Multi-Choice QA | Example Open-ended Generation |
|---|---|---|---|---|
| The Lord of the Rings | High Monthly Views: 177540 | Art and Entertainment | Question: "Who is the main protagonist of 'The Lord of the Rings'?", Options: A: "Frodo Baggins", B: "Gandalf the Grey", C: "Aragorn", D: "Legolas". Answer: A | Question: "Who is the author of 'The Lord of the Rings' trilogy?" Answer: "J.R.R. Tolkien." |
| Detritivore | Medium Monthly Views: 11810 | Biology | Question: "What do detritivores consume to obtain nutrients?", Options: A: "Fresh, living plants and animals", B: "Detritus, including decomposing plant and animal parts and feces", C: "Sunlight and water", D: "Inorganic minerals and metals". Answer: B | Question: "What term is used for the consumption of dead wood by detritivores?" Answer: "Sapro-xylophagy." |
| Maluma | Low Monthly Views: 2252 | Person | Question: "In which city was Maluma born and raised?" Options: A: "Bogotá", B: "Cali", C: "Medellín", D: "Cartagena". Answer: C | Question: "What is the name of Maluma's 2023 album?" Answer: "Don Juan" |

Table 5: Example data from the SpecWiki dataset.

## B.3 Hallucination Metric Descriptions

In this experiment, we additionally incorporate two metrics, Semantic Entropy (Kuhn et al., 2022) and Local Intrinsic Dimension (LID) (Yin et al., 2024), to assess hallucination. This helps evaluate whether the finetuning methods that enhance Parameter Specialization also effectively mitigate the unintended side effect of hallucination in the model's output.

**Semantic Entropy** Semantic entropy is defined as a measure of uncertainty based on the distribution of semantically equivalent outputs. In this method, the outputs are grouped into clusters of semantically similar responses, and the entropy is calculated among these groups. Formally, it is expressed as:

$$\text{Semantic Entropy} = \frac{1}{|C|} \sum_{i=1}^{|C|} \log p(C_i | x)$$

where $C_i$ represents the summed likelihood of outputs in the $i$-th group, and $|C|$ is the total number of such groups. The measure captures the uncertainty not in individual responses but within clusters of semantically similar outputs. This approach accounts for semantic equivalence among different responses, providing a more robust evaluation of entropy in generative tasks.

**Local Intrinsic Dimension** The Local Intrinsic Dimension (LID) method detects hallucinations in Large Language Models by measuring the discrepancy in the local intrinsic dimension of model activations. This approach is grounded in the principle that LID represents the minimal number of activations required to characterize a data point, with truthful outputs exhibiting lower LID values due to their closer alignment with natural language structure, while hallucinated outputs tend to show higher LID values due to mixing human prompt and model distributions. Technically, the method employs Maximum Likelihood Estimation (MLE) using a Poisson process to approximate the count of neighbors surrounding sample points, computed through the formula $m(X_i) = (1/(T - 1) * \sum(log(Q_T/Q_j)))^{-1}$, where T represents the number of nearest neighbors and $Q_j$ denotes the Euclidean distance to the j-th nearest neighbor. For more details about the Local Intrinsic Dimension metric, please refer to the work (Yin et al., 2024).

