# OpenReview forum: "The Rise of Parameter Specialization for Knowledge Storage in Large Language Models"
_NeurIPS.cc/2025/Conference — NeurIPS 2025 poster_

### Official Review · Reviewer_kanx · 2025-06-30

**Clarity:** 3
**Significance:** 2
**Originality:** 2
**Rating:** 4
**Confidence:** 3

**Summary:**

The authors analyze the MLP layers of 20 open-source LLMs to determine the relationship between performance and the way knowledge is stored. They build on the key-value interpretation by Geva et al. (2021) and inspect the absolute values of the intermediate representations between the up- and down-projection matrix in the MLP layers. In their emprical analysis, they authors use a newly developed knowledge benchmark (SpecWiki), based on Wikipedia, which covers knowledge concepts with varying frequencies. They define a new metric, called the Parameter Specialization Score (PSS) which measures the absolute difference between the model’s post-masking Concept-Specific Score (accuracy on questions about the target concept) and its General Score (accuracy on unrelated control questions) divided by the model’s original overall accuracy. The authors find that as language models become more advanced and demonstrate stronger knowledge capabilities, their parameters exhibit increased specialization, and the specialized distribution of knowledge contributes to improving the efficiency of knowledge utilization in LLMs.

**Questions:**

- Could you share a sensitivity sweep showing how PSS changes with different masking ratios (and any alternative denominator choices) so we can confirm the size–quality correlation isn’t just a scaling artefact?
- The authors discuss the concept of Knowledge Superposition in LLMs. Maybe it would be interesting to discuss how your metric relates, or may suffer, from the concepts of mono- and polysemanticity?

Bricken, et al., "Towards Monosemanticity: Decomposing Language Models With Dictionary Learning", Transformer Circuits Thread, 2023. (https://transformer-circuits.pub/2023/monosemantic-features/index.html)

**Ethical Concerns:**

["NO or VERY MINOR ethics concerns only"]

**Final Justification:**

The authors have responded to some of my concerns and overall I believe this paper could be a valuable contribution to NeurIPS. However, the limited originality compared to earlier work makes me keep the score of 4 (borderline accept).

**Limitations:**

The authors clearly describe the most important limitations of their work.

**Paper Formatting Concerns:**

No concerns.

**Quality:**

3

**Strengths And Weaknesses:**

Strenghts:
- The authors introduce the Parameter Specialization Score (PSS) and a new benchmark (SpecWiki) which is a reproducible and clear metric to quantify how narrowly facts are tied to individual MLP value vectors.
- The authors include 20 open source LLMs in their analysis which shows the positive relationship between factual accuracy and higher PSS.
- Showing that fine-tuning only the high-PSS vectors improves recall and lowers hallucinations is a lightweight way to regularize models.

Weaknesses:
- PSS divides the post-mask accuracy gap by the pre-mask control accuracy, so baseline model strength distorts the score. Apart from one diagnostic sweep in Figure 3, the paper reports only the aggregated PSS values which makes it impossible to inspect the underlying pre- and post-mask accuracies that would reveal which term really drives the correlation. We can’t know whether the size–quality trend is genuine or a scaling artefact.
- As the authors acknowledge in their limitations section: prior work shows attention heads, router experts and even layer-norm scales also store factual info (e.g., Geva et al., 2023).

---

> ### Author Rebuttal · Authors · 2025-07-31
>
> We sincerely thank Reviewer kanx for the thoughtful and encouraging feedback. We truly appreciate your recognition of our proposed Parameter Specialization Score metric, the design of the SpecWiki benchmark, and the significance of our large-scale analysis across 20 open-source LLMs. Given the valuable questions you raised, we have provided detailed responses below and would be happy to address any further questions or concerns the reviewer may have.
>
> > **W1 & Q1:** PSS divides the post-mask accuracy gap by the pre-mask control accuracy, so baseline model strength distorts the score. Apart from one diagnostic sweep in Figure 3, the paper reports only the aggregated PSS values which makes it impossible to inspect the underlying pre- and post-mask accuracies that would reveal which term really drives the correlation. We can’t know whether the size–quality trend is genuine or a scaling artefact... Could you share a sensitivity sweep showing how PSS changes with different masking ratios (and any alternative denominator choices) so we can confirm the size–quality correlation isn’t just a scaling artefact?
>
>
> Thank you for the great questions.
>
> To better address your concerns, we would like to further clarify and explain the formulation of the Parameter Specialization Score (PSS) we proposed.
>
> 1. Specifically, we first calculate the difference between the model’s average performance on the unrelated questions set and its average performance on the concept-related dataset, after ablating the located target parameters. **This difference represents how much of the knowledge contained in the located target parameters is both parameter-specific and concept-related.** Since each model performs differently on the SpecWiki dataset, we use its original (unintervened) performance as the denominator to serve as a proxy for the total knowledge encoded in the model. **This allows us to finally calculate what proportion of the concept-related, parameter-specific knowledge accounts for the model's overall knowledge.**
>
> 2. Suppose we do not include the denominator in the PSS formula. Consider two 7B models—one with stronger performance (e.g., achieving 80% accuracy on SpecWiki) and another with weaker performance (e.g., 40% accuracy). If we ablate the same 10% of MLP parameters, and both models lose the same proportion of parameter-specific, concept-related knowledge relative to their total knowledge (i.e., the drop in the numerator accounts for 20% of the denominator in both cases), the stronger model will show a larger absolute performance drop in the numerator (0.8 × 20% vs. 0.4 × 20%). This would make it appear to have a higher degree of parameter specialization, but in reality, it simply contains more target knowledge to begin with, as evidenced by its stronger original performance.
>
> Moreover, the results in Figure 2 show that stronger-performing models tend to exhibit higher Parameter Specialization Scores. If we remove the denominator—which currently offsets the advantage of stronger models—the trend in Figure 2 would appear even steeper and more pronounced.
>
>
> > **W2:** As the authors acknowledge in their limitations section: prior work shows attention heads, router experts and even layer-norm scales also store factual info (e.g., Geva et al., 2023).
>
> Although subsequent studies have claimed that other components within LLMs may also store factual knowledge (as we discuss in our Limitations section), we follow the prevailing perspective in the field of LLM interpretability, which posits that the MLP components are the primary parameters responsible for storing knowledge in the model. Furthermore, the view that value vectors within MLPs act as units of knowledge storage offers us a more practical way to investigate and quantify parameter specialization.
>
> Consequently, in our research on knowledge storage patterns, we concentrate our investigation on the MLP components. In future work, we will extend our analysis to include a broader range of parameters across other components of the transformers.
>
> > **Q2:** The authors discuss the concept of Knowledge Superposition in LLMs. Maybe it would be interesting to discuss how your metric relates, or may suffer, from the concepts of mono- and polysemanticity?
>
> Thank you for pointing out the important concept of Knowledge Superposition again.
>
> The knowledge superposition discussed in this paper **is indeed related to the concepts of mono- and polysemanticity.** For example, the word "light" can mean both **"not heavy" and "illumination."** Although these meanings are associated with the same token, we believe that the knowledge parameters representing the "not heavy" meaning should be grouped more closely with other parameters that also represent the concept of lightness in weight, while the parameters representing "illumination" should be closer to those responsible for knowledge related to light or darkness. **This corresponds to a lower degree of knowledge superposition and greater parameter specialization.**
>
> According to the experiments and conclusions of this paper, such an arrangement allows the model to retrieve the target knowledge more efficiently and reduces hallucinations.

---

### Official Review · Reviewer_7hsq · 2025-07-01

**Clarity:** 3
**Significance:** 2
**Originality:** 3
**Rating:** 3
**Confidence:** 3

**Summary:**

This paper investigates how large language models store knowledge within their MLP parameters, analyzing twenty open-source models. The findings indicate that more advanced models with stronger knowledge capabilities exhibit increased specialization in their MLP parameters, meaning parameters are more focused on encoding similar types of knowledge. This specialized knowledge distribution is shown to improve the efficiency of knowledge utilization.

**Questions:**

Why is 0 used in Formula 4 instead of random noise?

**Ethical Concerns:**

["NO or VERY MINOR ethics concerns only"]

**Final Justification:**

The rebuttal addresses most of my concerns, and I will raise the score from 2 to 3.

**Limitations:**

yes

**Quality:**

2

**Strengths And Weaknesses:**

Strengths：
The paper conducts extensive experiments to validate the relationship between parameter size and knowledge.
Weaknesses：
The paper doesn't propose a new method to find the relationship between knowledge and parameters; it directly uses existing methods. Furthermore, the perspective for examining parameters is singular, relying solely on Causal Tracing. Other methods like Gradient Attribution or statistical methods [1] were not considered.
The paper's conclusion lacks robustness, as there's insufficient evidence to support the SpecWiki Performance range of 0.4-0.7. This makes it difficult to establish a consistent pattern.
The data construction carries a risk of noise. Randomly selecting knowledge from Wikipedia pages over just one year means it could have been modified later, leading to discrepancies between this knowledge and the real world or the model's understanding.
The paper's statements are inaccurate. It only considers MLP layer parameters, so it cannot be generalized as a relationship between model size and knowledge.
[1] Knowledge neurons in pretrained transformers

---

> ### Author Rebuttal · Authors · 2025-07-31
>
> We sincerely thank Reviewer 7hsq for the thoughtful feedback and for acknowledging the strength of our experimental validation. We believe, however, that some concerns stem from misunderstandings about the scope and goals of our work. In the following response, we clarify these points in detail and aim to further highlight the core contributions of our paper. We would be happy to respond to any further questions or concerns the reviewer may have.
>
>
> > **W1:** The paper doesn't propose a new method to find the relationship between knowledge and parameters; it directly uses existing methods.
>
> The key contribution of our work lies in introducing a novel perspective and metric for understanding the storage pattern of knowledge within LLM parameters—specifically, the degree to which certain parameters are **specialized** or **dedicated** to a particular knowledge. Our goal is not to propose a new intervention, attribution, or visualization method for measuring the relationship between parameters and knowledge. The existing interpretability methods we used are sufficient to support the new perspective proposed in this work.
>
> > **W2:** Furthermore, the perspective for examining parameters is singular, relying solely on Causal Tracing. Other methods like Gradient Attribution or statistical methods were not considered.
>
> As we mentioned in W1, the interpretability methods used in this work are not the focus of innovation or contributions. Instead, our aim is to apply established and mature interpretability techniques to uncover new patterns and phenomena between model parameters and knowledge, and to demonstrate that these phenomena help the model better leverage the target knowledge (Section 5).
>
> Moreover, causal tracing has already become a widely adopted and mainstream interpretability method for verifying the correlation between model parameters and target knowledge [1, 2, 3], and our use of it follows these prior studies.
>
> > **W3:** The paper's conclusion lacks robustness, as there's insufficient evidence to support the SpecWiki Performance range of 0.4-0.7. This makes it difficult to establish a consistent pattern.
>
> SpecWiki is a dataset we propose, consisting of 525 Wikipedia concept entries with varying frequencies. For each concept, we design 10 distinct questions in two different formats. The full construction process of SpecWiki has been described in detail in Section 3; it is both well-justified and fully reproducible. And we have already released the complete dataset in the Supplementary Material. Moreover, anyone can follow our methodology to construct a similar dataset in their own domain, ensuring that parameter specialization across models remains consistent and generalizable.
>
> Please feel free to raise any further questions or points of confusion. We sincerely appreciate your engagement in the discussion and would be glad to know if our response has addressed your concerns and clarified the contributions of our work.
>
>
> > **W4:** The data construction carries a risk of noise. Randomly selecting knowledge from Wikipedia pages over just one year means it could have been modified later, leading to discrepancies between this knowledge and the real world or the model's understanding.
>
> 1. **It is important to note that we did not simply choose the year "2019."** In fact, the 2019 version of Wikipedia does not only include concepts newly introduced in that year, **but also encompasses all the knowledge entries accumulated over the previous years.** As such, its content can be seen as an approximation of the majority of knowledge concepts available to human society up to 2019. Based on this vast distribution of concepts, we performed random sampling and ultimately obtained 525 entries with varying frequencies, which constitute our SpecWiki dataset.
>
> 2. Secondly, 2019 is already considered relatively recent for the pretraining corpora of most models. As described in Section 3.4, for example, the pretraining dataset of GPT-J only includes data up to and including 2019. Therefore, even though some knowledge has been updated in the years since, the proportion of such updates in the data would be minimal.
>
> > **W5:** The paper's statements are inaccurate. It only considers MLP layer parameters, so it cannot be generalized as a relationship between model size and knowledge.
>
> We adopt the prevailing perspective in the field of LLM interpretability [1, 2, 4], which posits that the MLP components are the primary parameters responsible for storing knowledge in the model, while the attention components are more focused on transporting that knowledge, and other components serve additional distinct roles. These points have been discussed in our Limitations section — Section A in the Appendix. Furthermore, the view that value vectors within MLPs act as units of knowledge storage [2, 4] provides us with a more practical way for investigating and quantifying parameter specialization across models.
>
> Consequently, in our study of knowledge storage patterns, we focus our analysis on the MLP components, which we believe is sufficient for current efforts in quantifying factual knowledge in LLMs. In future work, we will expand our analysis to include a broader range of parameters from other components within the transformers.
>
>
> > **Q1:** Why is 0 used in Formula 4 instead of random noise?
>
> Let us consider a hypothetical scenario: suppose there are ten value vectors, each contributing equally to the representation of Harry Potter domain knowledge, with their feature directions aligned and coefficients all set to 0.1 (see Equation 2 for the definition of coefficients). If we reduce the influence of one of these value vectors (i.e., set its coefficient to zero), the representation of Harry Potter knowledge remains unaffected due to the presence of nine redundant carriers of the same information. This aligns with the realistic expectation of robustness in knowledge representation.
>
> However, if we instead add a noise vector of the same magnitude but in a random direction (which is likely orthogonal to the original target direction [1]) to that value vector—even with a scale as small as 0.1 (a commonly used scale in recent intervention studies [2, 3])—the resulting direction can deviate significantly. This may lead to a notable distortion in the overall representation, falsely suggesting a change in the importance of the targeted knowledge.
>
> We therefore believe that **ablation (or “knock-out”)** provides a more accurate means of assessing the contribution of specific parameters to a given piece of knowledge, compared to **noise-based interventions**.
>
>
> —
>
> **References:**
>
> [1] Locating and Editing Factual Associations in GPT
>
> [2] Transformer Feed-Forward Layers Are Key-Value Memories
>
> [3] Open Problems in Mechanistic Interpretability
>
> [4] Transformer Feed-Forward Layers Build Predictions by Promoting Concepts in the Vocabulary Space

---

> > ### Comment · Reviewer_7hsq · 2025-08-07
> >
> > Thank the authors for the rebuttal. It addresses most of my concerns. I will raise the score.

---

### Official Review · Reviewer_BzMR · 2025-07-02

**Clarity:** 4
**Significance:** 3
**Originality:** 3
**Rating:** 4
**Confidence:** 3

**Summary:**

This paper investigates how large language models store and utilize knowledge within their MLP parameters, revealing that stronger models increasingly specialize their parameters to encode similar types of knowledge more compactly, while weaker models distribute knowledge more diffusely. By analyzing 20 open-source LLMs with a new benchmark, and validating through controlled fine-tuning experiments, the authors demonstrate that higher parameter specialization improves knowledge retrieval efficiency, reduces hallucinations, and boosts performance on knowledge-intensive tasks. These findings highlight the importance of aligning parameter structures with knowledge retrieval mechanisms to achieve both greater interpretability and better performance in LLMs.

**Questions:**

1.This formula 6 can measure parameter specialization, but why is it normalized in this way?

2.In Figure 3, why is the vertical axis (the score difference between the two) zero when the mask rating is 0? Does that mean their accuracies are equal?

**Ethical Concerns:**

["NO or VERY MINOR ethics concerns only"]

**Final Justification:**

I have carefully reviewed the authors' rebuttal and my concerns are mostly addressed in the rebuttal. The score will remain positive.

**Limitations:**

Yes

**Quality:**

3

**Strengths And Weaknesses:**

Strengths

1.This work systematically quantify and compare the degree of parameter specialization across different large language models, proposing the Parameter Specialization Score (PSS) as a metric and validating its effectiveness through multiple experimental methods.

2.The paper conduct large-scale and systematic experiment, covering 20 open-source large language models with a carefully constructed benchmark, as well as its comprehensive parameter-masking and fine-tuning experiments. These extensive experiments strongly support and validate the paper’s conclusions about parameter specialization, ensuring their correctness and demonstrating the practical relevance of the findings for understanding and improving knowledge storage in LLMs.

Weaknesses

The conclusion is relatively narrow. although the paper conducts extensive experiments, its findings mainly describe observed model phenomena, offering limited actionable insights.

---

> ### Author Rebuttal · Authors · 2025-07-31
>
> We sincerely thank Reviewer BzMR for the detailed and encouraging feedback. We greatly appreciate your recognition of our efforts to systematically quantify parameter specialization across LLMs, the introduction of the Parameter Specialization Score (PSS), and the validation of our findings through comprehensive experiments. We have addressed each of the valuable questions you raised in the responses below, and we would be happy to respond to any further questions or concerns you may have.
>
> > **W1:** The conclusion is relatively narrow. although the paper conducts extensive experiments, its findings mainly describe observed model phenomena, offering limited actionable insights.
>
> Thank you very much for raising this great point.
>
> Our work is the first to propose and investigate the concept of **Parameter Specialization**, and we quantify how model parameters specialize in specific knowledge across different series and scales of LLMs. We not only show that this trend contributes positively to a model’s ability to utilize target knowledge (as demonstrated in the experiments in Section 5), but also highlight that this direction holds substantial potential for further research. It deserves more attention and deeper exploration from the research community — for example: *Given the constraints of a fixed-size pre-training dataset, how can we better organize the distribution of knowledge data across a model's internal parameters to improve knowledge utilization efficiency?*
>
> Here, we propose two actionable research directions:
>
> 1. As demonstrated in Section 5, through extensive controlled and contrastive fine-tuning experiments, we have shown that fine-tuning only the parameters responsible for the target knowledge—while freezing other unrelated parameters—can help the model better acquire the target knowledge, while minimizing negative interference with unrelated capabilities. Therefore, future work could explore more precise methods for localizing model parameters that are responsible for specific target knowledge, as well as develop more efficient and robust techniques for sparse fine-tuning.
>
> 2. During pretraining, organizing knowledge-related data fragments close together in the corpus—such as grouping Harry Potter novels, related articles, and corresponding Wikipedia entries—may promote greater parameter specialization. Based on the findings in [1, 2], we hypothesize that this approach facilitates the backpropagation of gradients associated with shared knowledge into a concentrated set of parameters, thereby enhancing the overall degree of parameter specialization.
>
>
> Our work opens up a new interpretability perspective for understanding knowledge storage in LLMs—an area that remains underexplored and calls for further investigation by the research community. In the revised version of the paper, we will include a dedicated subsection discussing these potential future work in more detail.
>
>
> > **Q1:** This formula 6 can measure parameter specialization, but why is it normalized in this way?
>
> Thank you for the good questions.
>
> We define this formula to quantify the extent of overlap and superposition between the concept-related knowledge embedded in the target parameters, and the other unrelated knowledge.
>
> Specifically, we first calculate the difference between the model’s average performance on the unrelated questions set and its average performance on the concept-related dataset, after ablating the located target parameters. **This difference represents how much of the knowledge contained in the located target parameters is both parameter-specific and concept-related.** Since each model performs differently on the SpecWiki dataset, we use its original (unintervened) performance as the denominator to serve as a proxy for the total knowledge encoded in the model. **This allows us to finally estimate what proportion of the concept-related, parameter-specific knowledge accounts for the model's overall knowledge.**
>
> If the PSS (Parameter Specialization Score) is closer to 1, it indicates a higher proportion of concept-related knowledge and a lower proportion of unrelated knowledge in the target parameters. Conversely, a lower score suggests that the target parameters contain less concept-related knowledge and more unrelated knowledge.
>
> > **Q2:** In Figure 3, why is the vertical axis (the score difference between the two) zero when the mask rating is 0? Does that mean their accuracies are equal?
>
> Yes.
>
> Since our benchmark evaluation runs through all the concepts in the dataset and averages the model’s performance across them, to compute the Concept-Specific score (y-axis), a masking rating of 0 means that we do not intervene in the model’s internal inference in any way during this process.
>
> In this case, the model’s performance on concept-specific questions across all concepts is identical to its average accuracy on general questions (which are also evaluated across all concepts in the entire benchmark).
>
>
>
> ---
>
> **References**：
>
> [1] Backward Lens: Projecting Language Model Gradients into the Vocabulary Space
>
> [2] How Do Large Language Models Acquire Factual Knowledge During Pretraining?

---

> > ### Comment · Reviewer_BzMR · 2025-08-06
> >
> > Thank you for the authors' reply. The authors explained my question, but I think that further experimental improvements can be made in the application of the discovery. I have decided to keep my score unchanged.

---

> > > ### Author Response · Authors · 2025-08-07
> > > **Responses to Reviewer BzMR**
> > >
> > > Thank you Reviewer BzMR for the prompt response, and for your recognition of the current contributions of our paper!
> > >
> > > We believe that the proposed notion of Parameter Specialization, along with our analysis and empirical results, already constitutes a substantial contribution. Adding further application-oriented studies would be somewhat beyond the scope of this work. However, we truly appreciate your suggestion — it is indeed a valuable direction for future exploration. We plan to continue investigating the potential applications of Parameter Specialization in model training in our follow-up work.

---

### Official Review · Reviewer_8bPy · 2025-07-03

**Clarity:** 4
**Significance:** 4
**Originality:** 3
**Rating:** 5
**Confidence:** 4

**Summary:**

The authors build on previous work showing that MLP layers in LLMs seem to function as key-value stores for persistent memories. They analyze the distribution of these memories over 20 LLMs, with a specific interest in the specialization of parameters for particular concepts. They find that stronger models exhibit higher parameter specialization, and weaker models distribute knowledge more diffusely across parameters.

This is performed by masking out keys relating to particular concepts based on the difference in weights for concept-related and irrelevant questions (the irrelevant questions prevents masking out “general abilities” like processing text for question answering). They measure specialization as “score on general tasks after surgery” minus “score of concept-specific tasks after surgery”, normalized by “score on general tasks before surgery”

**Questions:**

I would appreciate if the analysis were repeated with a "parameter specialization" score that more purely reflects specialization (as discussed in "weaknesses" above).

**Ethical Concerns:**

["NO or VERY MINOR ethics concerns only"]

**Final Justification:**

Concerns were suitably addressed by the authors. Score has been raised from 4 to 5.

**Limitations:**

yes, excepting the question raised above.

**Quality:**

3

**Strengths And Weaknesses:**

Strengths
- This is a very interesting finding, and shines some light on the question of trying explain mechanistically what happens within models as they get stronger. This is a compelling set of results indicating that parameter specialization in MLPs is one factor in model performance.
- I appreciate the novelty (at least to me) of the methods, e.g. applying the same analysis to different models within the same model family, or their method for applying concept-related and irrelevant questions to the model.

Weaknesses
- This is not a causal analysis – it is very possible that the strength of models does not \*arise\* from parameter specialization, but rather that parameter specialization is just a side effect of how stronger models are trained.
- I believe that the PSS score conflates two effects: the specialization of the parameters for a given task, and also how distributed a particular task is across multiple neurons. For example, you could have neurons that are each highly specialized for task X (they do not represent any task), but the task representation could be highly redundant such that even ablating the top few neurons for task X does not significantly affect performance on task X

---

> ### Author Rebuttal · Authors · 2025-07-31
>
> We sincerely thank Reviewer 8bPy for the thoughtful and encouraging feedback. We truly appreciate your recognition of the novelty of our methodology and the significance of our findings regarding parameter specialization. We have addressed each of the valuable questions you raised in the responses below, and we would be happy to respond to any further questions or concerns the reviewer may have.
>
> > **W1:** This is not a causal analysis – it is very possible that the strength of models does not *arise* from parameter specialization, but rather that parameter specialization is just a side effect of how stronger models are trained.
>
> Thank you for the question.
>
> We acknowledge that the degree of parameter specialization in a model does not have a perfect one-to-one correspondence with overall model performance. Model performance is influenced by many other factors, such as architecture, the quality and quantity of the pre-training data, and more.
>
> However, as shown in the multiple controlled and contrastive fine-tuning experiments in Section 5—where we injected an equal amount of knowledge into the same LLM by fine-tuning different groups of model parameters—we found that the approach of fine-tuning only the parameters responsible for the target knowledge, while freezing all unrelated parameters (i.e., increasing the degree of parameter specialization), consistently led to more efficient knowledge embedding. This includes better performance, reduced hallucination, and improved utilization of the target knowledge (see Table 2). In contrast, other approaches that did not further enhance parameter specialization also failed to achieve better model performance.
>
> Therefore, we provide evidence that increasing the degree of parameter specialization **causally improves** the model's ability to utilize target knowledge more effectively.
>
>
> > **W2&Q1:** I believe that the PSS score conflates two effects: the specialization of the parameters for a given task, and also how distributed a particular task is across multiple neurons. For example, you could have neurons that are each highly specialized for task X (they do not represent any task), but the task representation could be highly redundant such that even ablating the top few neurons for task X does not significantly affect performance on task X
>
> Thank you for the great question!
>
> We acknowledge the possibility that, in addition to a small subset of highly specialized parameters that contain the target knowledge, there may also exist other parameters that redundantly store this knowledge in a more dispersed manner. However, because our ablations in Section 4.1 are conducted over significant portions of the model’s parameters (10%, 20%, 30%, etc.), **we are ultimately investigating parameter specialization at the model-wide level**—not just at the level of individual specialized parameters, with the goal of comparing the knowledge storage patterns across different models.

---

> > ### Comment · Reviewer_8bPy · 2025-08-02
> >
> > Thank you for your thoughtful responses.
> >
> > Q1. I should have been clearer -- I believe that the causal mechanisms at pretraining may be quite different from those in finetuning. However, I don't think this is a major weakness for the paper.
> >
> > Q2. To clarify, my criticism of the PS5 score is that it conflates two different phenomena, and so it is unclear which of those two it is measuring. I think that the interpretation of the results will be clearer if the score is adjusted so that it measures one or the other!

---

> ### Author Response · Authors · 2025-08-06
> **Responses to Reviewer 8bPy (1/3)**
>
> We truly appreciate your prompt response and the additional clarification! We’d be happy to further address your concerns, and please feel free to let us know if anything remains unclear. **As for your second question, that’s really a great point. So we have made additional efforts here specifically to help address this issue.**
>
> > *Q1. I should have been clearer -- I believe that the causal mechanisms at pretraining may be quite different from those in finetuning. However, I don't think this is a major weakness for the paper.*
>
>
> 1. Firstly, due to limitations in computational resources, it is difficult for us to replicate and manually adjust the pretraining process, in order to study the causal impact of intentionally increasing parameter specialization on performance improvement. Therefore, we chose to use finetuning, which requires less computational resources and is more widely adopted—to investigate this causality.
>
> 2. According to previous work [1, 2, 3], finetuning has become an important approach both for continuously strengthening a model’s existing knowledge [1, 2] and for trying to incorporate new knowledge [3]. This aligns with our goal of enhancing the model’s efficiency in utilizing prior knowledge. Therefore, we draw upon these insights and design controlled finetuning experiments to investigate the causal relationship in question.
>
> While we acknowledge that pretraining and finetuning may operate under different mechanisms, this distinction does not compromise our central argument: further improving the degree of parameter specialization can improve model’s efficiency in utilizing target knowledge. Even if this improvement is causally achieved through finetuning, it reveals promising directions for pushing the upper bounds of model performance. In future work, **we will explore how parameter specialization can be intentionally increased during the pretraining process as well.** Such efforts would enable a deeper causal analysis and provide stronger evidence for its potential application in future pretraining strategies.
>
>
> ---
>
> **References:**
>
> [1] Learning Dynamics of LLM Finetuning
>
> [2] The Ultimate Guide to Fine-Tuning LLMs from Basics to Breakthroughs: An Exhaustive Review of Technologies, Research, Best Practices, Applied Research Challenges and Opportunities
>
> [3] Fine-Tuning or Retrieval? Comparing Knowledge Injection in LLMs

---

> > ### Comment · Reviewer_8bPy · 2025-08-06
> >
> > Thank you for your thoughtful response! I would appreciate if the acknowledgement of this limitation is added to the paper! Again, I don't think it's a significant weakness of the paper. I just think that the paper should be cautious in the stated interpretation of those results.

---

> > > ### Author Response · Authors · 2025-08-06
> > > **Thank you Reviewer 8bPy for the prompt response!**
> > >
> > > Thank you very much for the prompt and positive response, as well as the recognition of our paper's contributions!
> > >
> > > We sincerely appreciate your suggestion, and we will clarify the relevant wording in the paper accordingly and carefully. And additionally, we will include a more detailed discussion in the limitations section about the fine-tuning method part and the phenomenon of Knowledge Redundancy.

---

> ### Author Response · Authors · 2025-08-06
> **Responses to Reviewer 8bPy (2/3)**
>
> > *Q2. To clarify, my criticism of the PS5 score is that it conflates two different phenomena, and so it is unclear which of those two it is measuring. I think that the interpretation of the results will be clearer if the score is adjusted so that it measures one or the other!*
>
> Thank you once again for raising the important point regarding **Knowledge Redundancy**!
>
> After careful consideration and multiple rounds of reflection, we still believe that—even accounting for the existence of knowledge redundancy—our current formulation of PSS (Equation 6) remains the most appropriate and effective metric available for measuring the degree of parameter specialization with respect to target knowledge within the model. **Our reasoning is as follows:**
>
> For example, consider two target concepts, **Concept A** and **Concept B**, which encompass knowledge sets {a1, a2, a3, …, an} and {b1, b2, b3, …, bn}, respectively. Suppose **vector *a*** encodes the knowledge set {a1, a2, a3, …, an}, while **vector *b*** encodes the set {a1, b1, b2, …, bn} — this indicates that the knowledge piece **a1** is redundantly represented in both vectors.
>
> As a result, when **vector *a*** is ablated and the model is still able to answer questions related to a1, it suggests that the **vector *b*** has also been activated and invoked to retrieve the knowledge a1. **However, doing so will introduce unrelated knowledge from vector b, potentially leading to additional hallucinations**. Therefore, although **vector *a*** exclusively encodes knowledge related to Concept A, **we would still not consider this a well-specialized distribution of Concept A's knowledge within the model.**
>
> Now, consider computing the PSS score by ablating each vector individually:
>
> - For **Concept A**:
>   `PSS_A = (1 - 1/n) / (2n / 2n) = 1 - 1/n`
>
> - For **Concept B**:
>   `PSS_B = (1 - 0) / (2n / 2n) = 1`
>
> where n denotes the number of knowledge pieces within each concept.
>
> Even though **vector b** also encodes some unrelated knowledge from Concept A (i.e., a1), we believe this still represents the **best achievable specialization** for Concept B within the model.
> **Therefore, we believe that the computed PSS scores above in this scenario align well with our intended interpretation of specialization for Concepts A and B** (that B achieves a better degree of parameter specialization in the model). Ideally, we believe redundant knowledge pieces (like **a1**) that were distributed across unrelated vectors, should instead be concentrated in the vectors associated with their respective target concepts. Otherwise, activating such redundant knowledge in the unrelated vectors may also bring in irrelevant information, increasing the risk of hallucinations and impairing the model’s performance on the intended knowledge.
>
> Therefore, we would like to clarify: **our definition of parameter specialization does not refer to the specialization of any single parameter toward the target knowledge, but rather to the overall distribution of that knowledge across model parameters.** In our view, the more concentrated the knowledge is—i.e., controlled by fewer parameters—the more easily it can be accessed or utilized by the model. We believe the PSS score we propose now accurately captures this notion of specialization.
>
> ---
>
> **We sincerely appreciate you raising this thoughtful question**, which has helped us further improve this paper. We will revise the corresponding wording in the revised version to reflect this point more precisely.

---

> > ### Comment · Reviewer_8bPy · 2025-08-06
> >
> > Imagine this scenario: there M concepts, and for each concept are N vectors {v1, ... vN} that are all perfectly specialized for that concept. There are no other vectors in the model. But N is greater than k (the number of elements we allow in S_l). In this case, the current score would show low "specialization", even though this scenario is perfectly specialized.
> >
> > I think this case (and others) show that it is difficult to interpret the current instantiation of the PSS metric.
> >
> > I sincerely think that this is a very interesting and promising paper. I think that it just needs cleanup of the main metric so that the interpretation of the results can be clearer. I really hope that the authors will take this last 10% jump to turn this into a great paper!
> >
> > On a very minor note, in future versions of this paper, I would also love to see some discussion of existing work on disentanglement (e.g. beta-VAEs) and also mixture-of-experts, to understand how this work fits in with that research.

---

> ### Author Response · Authors · 2025-08-06
> **Responses to Reviewer 8bPy (3/3)**
>
> ### **Optional Reading**:
>
> When we investigated the point about **Knowledge Redundancy**, we also conducted additional experiments to verify whether knowledge redundancy truly exists in the model and to what extent. And we will include these results in the appendix.
>
> ### **Zero-Ablation vs. Gaussian Noise Intervention experiment:**
>
> Specifically, following the procedure outlined in Section 3.2 and according to Equations 2 and 5, we compute S^ℓ and sort the results in descending order. In this experiment, we set k = 5 and 200 of the total number of value vectors, meaning we intervene on only the top k value vectors in the MLP of each layer. For example, when k = 5, this corresponds to less than 0.05% of all value vectors in the MLPs of a 7B model. This time, we apply two types of interventions:
>
> - One is the same **zero-ablation** method as before (following Equation 4),
>
> - The other is **Gaussian noise intervention**, where we add random Gaussian noise with a scale of 0.1 to the selected value vectors—**without changing their corresponding coefficients** in Equation 4. This noise scale is consistent with previous studies on value vectors in MLPs [1, 2].
>
> We apply both interventions across 6 models, and the results are shown below:
>
> | Model         | Performance Without Intervention | Zero-Ablation - Top 5 vectors (< 0.05%) | Zero-Ablation - Top 500 vectors (< 5%) | Gaussian Noise - Top 5 vectors (< 0.05%) | Gaussian Noise - Top 500 vectors (< 5%) |
> |---------------|----------------------|------------------------------------------|----------------------------------------|-------------------------------------------|-------------------------------------------|
> | LLaMA-7B     | 0.53                 | 0.52 vs 0.53                              | 0.39 vs 0.50                            | 0.40 vs 0.51                               | 0.38 vs 0.49                               |
> | LLaMA2-7B     | 0.60                 | 0.60 vs 0.60                              | 0.40 vs 0.56                            | 0.42 vs 0.57                               | 0.39 vs 0.55                               |
> | LLaMA3-7B     | 0.85                 | 0.84 vs 0.85                              | 0.53 vs 0.76                            | 0.56 vs 0.82                               | 0.50 vs 0.75                               |
> | Qwen1.5-7B    | 0.72                 | 0.71 vs 0.72                              | 0.50 vs 0.65                            | 0.53 vs 0.68                               | 0.51 vs 0.64                               |
> | Qwen2.5-7B    | 0.81                 | 0.81 vs 0.81                              | 0.47 vs 0.73                            | 0.50 vs 0.78                               | 0.46 vs 0.71                               |
> | Qwen3-8B      | 0.88                 | 0.88 vs 0.88                              | 0.49 vs 0.74                            | 0.51 vs 0.85                               | 0.48 vs 0.71                               |
>
> Each cell shows Concept-Specific Score vs. Unrelated Knowledge Score. From the table, we observe that:
>
> 1. When applying zero-ablation to only top 5 value vectors, the model’s performance on concept-specific knowledge tasks remains largely unaffected; Similarly, performance on non-related tasks also remains stable. However, when applying Gaussian noise to the exact same 5 value vectors, performance on concept-specific knowledge drops significantly, while unrelated knowledge tasks are only minimally impacted. These findings indicate that **knowledge redundancy does indeed exist within the model’s parameters**: simply zero-ablating a small number of the most influential value vectors does not drastically impact the model’s expression of specific knowledge, due to redundant information present in other parts of the model.
>
> 2. At the same time, the top 5 selected value vectors are also indeed **highly activated and highly concept-specific**: Their high activation causes them to significantly affect the representation of target knowledge when perturbed with **Gaussian noise** (as shown by the significant drop in the Concept-Specific Score), despite the existence of redundancy. Their concept-specificity means they carry relatively little unrelated information, so corrupting them with noise has minimal effect on non-target knowledge performance (as shown by the minimal change in the Unrelated Knowledge Score).
>
> 3. When the number of intervened vectors is increased to 500 (< 5% of the total vectors), the impact on Concept-Specific Score becomes similar for both zero-ablation and noise intervention. This suggests that **as the intervention scale grows, the influence of knowledge redundancy within the parameters gradually diminishes. (This is also why we chose 10%, 20%, 30%, etc., as the intervention scales in the main experiments.)**
> ---
> **References**:
>
> [1] Transformer Feed-Forward Layers Are Key-Value Memories
>
> [2] Open Problems in Mechanistic Interpretability

---

> ### Author Response · Authors · 2025-08-07
> **Thank you for the great suggestions!**
>
> Thank you very much for the valuable suggestions and for proposing additional boundary conditions!
>
> 1. We greatly appreciate your mention of this boundary situation. Given the vast pretraining corpora of current LLMs and the **huge** number of concepts stored within the model, it can be reasonably inferred that the number of value vectors required for the perfect specialization of any single concept would constitute **far less than 10% of the total MLP value vectors**. And as shown in Section 4.1 of our main content, our selected values of k (e.g., 10%, 20%, 30%) cover a number of vectors that is certainly much larger than what is needed for the specialization of any single concept.
>
> However, we truly appreciate this insightful question, and we will include more discussion on this point and consider more boundary cases in the revised version of the paper to help clarify and validate our formulation.
>
> 2. **Regarding disentanglement and MOE**: We sincerely thank you for drawing a connection between our proposed Parameter Specialization and these novel and advanced notions or architectures. **This also helps demonstrate the broader potential applicability of our idea.** We will include a more detailed discussion on this topic in the updated version of the paper.
>
> Thank you once again for your strong recognition and appreciation of our work!

---

### Official Review · Reviewer_vpb2 · 2025-07-03

**Clarity:** 3
**Significance:** 3
**Originality:** 3
**Rating:** 5
**Confidence:** 4

**Summary:**

This paper analyzes knowledge storage in LLMs from a novel perspective: the specialization of knowledge storage. A higher specialization means that the knowledge of similar concepts is stored in a smaller range of neurons, instead of being spread out across many neurons. This paper leverages the idea that knowledge is stored as key-value pairs in the MLP layers. It identifies the specific values stored in the MLP layers related to specific concepts, which could indicate the degree of knowledge specialization. The experiments discover that increased knowledge specialization indicates stronger model performance.  Then, the paper utilizes the findings to improve fine-tuning. It finds that utilizing the specialized distribution of knowledge contributes to improving the efficiency of knowledge utilization.

**Questions:**

Please refer to the weaknesses.

**Ethical Concerns:**

["NO or VERY MINOR ethics concerns only"]

**Final Justification:**

The author's rebuttal addressed most of my concerns. I think the paper is generally solid and novel, and it has interesting findings which might benefit future research. It has some minor points to improve, such as the ablation method I mentioned, or the concern of the PSS score mentioned by other reviewers, but I think these points are not big issues, and this paper's contributions worth a positive score. So I decide to maintain my score (5).

**Limitations:**

yes

**Quality:**

4

**Strengths And Weaknesses:**

### Strengths:

(1) The motivation is novel and meaningful. It is meaningful to study knowledge storage in LLMs, but previous papers usually study knowledge storage locations, knowledge editing, etc. This paper takes a novel perspective: the knowledge specialization. The findings might benefit future applications in knowledge updating, LLM fine-tuning, etc.

(2) The proposed methods are reasonable. This paper mainly relies on the idea that the MLP layers act as key-value pairs to store knowledge. It identifies the “values” in the MLP layers that are related to specific concepts. Then, we can mask out these value vectors to evaluate the influence. It is a good method.

(3) The paper covers a thorough study cycle, from finding the positive relationship between knowledge specialization and model performance, to analyzing other related factors such as how parameter specialization develops during pretraining, to applying the findings to improve the fine-tuning.

(4) This paper conducts experiments on 20 models, which is very extensive. It also conducts extensive ablation studies.

(5) The experiment results well support the findings.

(6) The writing is clear and easy to follow.

### Weaknesses:

(1) The masking method uses zero-ablation, which means to set the weight of the masked value vector to zero. It might change the output scale, which might cause other interference. It might be better to adjust other value vectors’ weights to ensure the total weight is not changed.

(2) Lines 213-214 mention that the fixed k values are 10%, 20%, 30%, 40%, and 50%. I am not sure, but it might be possible that some concepts are only stored in very few neurons. If we take at least 10% of neurons, it might be still too many for those concepts.

(3) It would be better if the authors could provide some insights into how the findings can benefit real-world applications. For example, how to more effectively train a LLM, or some recommendations on knowledge updates, etc.

---

> ### Author Rebuttal · Authors · 2025-07-31
>
> We sincerely thank Reviewer vpb2 for the strong and encouraging recognition of our work — including the recognition of the novelty and significance of our motivation and ideas, the rigor and thoroughness of our methodology, the completeness and strength of our experimental results in supporting our conclusions, as well as the clarity of our writing and structure!
>
> We also greatly appreciate the valuable suggestions and insightful questions raised. We address these points in detail below and would be happy to respond to any further questions or concerns the reviewer may have.
>
> > **W1:** The masking method uses zero-ablation, which means to set the weight of the masked value vector to zero. It might change the output scale, which might cause other interference. It might be better to adjust other value vectors’ weights to ensure the total weight is not changed.
>
> Thank you for pointing this.
>
> However, we believe that keeping the total weight unchanged by scaling up the coefficients of unrelated value vectors in the MLP—those not selected—**could push the model's representations in unintended directions and introduce spurious importance**. Given this concern, we argue that maintaining zero-ablation, which isolates a single variable, is a more accurate and reliable choice for measuring the influence of the located parameters and assessing their degree of parameter specialization.
>
> > **W2**: Lines 213-214 mention that the fixed k values are 10%, 20%, 30%, 40%, and 50%. I am not sure, but it might be possible that some concepts are only stored in very few neurons. If we take at least 10% of neurons, it might be still too many for those concepts.
> Thank you for pointing these out.
>
> In our experiments, we observed that even though parameter specialization does occur for the vast majority of concepts, the overall knowledge distribution remains relatively spread out. For example, on our SpecWiki dataset, **we have tested all concepts** and found that ablating only 10% of the value vectors in the MLPs (by setting their corresponding coefficient values to zero) is generally insufficient to reduce performance to less than half of its original level.
>
>
> > **W3:** It would be better if the authors could provide some insights into how the findings can benefit real-world applications. For example, how to more effectively train a LLM, or some recommendations on knowledge updates, etc.
>
> In this work, we have demonstrated that the trend of parameter specialization in knowledge contributes positively to a model’s ability to utilize target knowledge (as shown in the experiments in Section 5). Specifically, it enables more effective retrieval of the target knowledge, reduces interference from irrelevant information, and ultimately contributes to enhanced accuracy and a decrease in hallucinations.
>
> These findings offer important insights into a key research question: Given the constraints of a fixed-size pre-training dataset, how can we better organize the distribution of these knowledge data across the model's internal parameters?
>
> Here, we propose two actionable research directions:
>
> 1. **Sparse Finetuning in the Post-Training Phase**
>
>    As demonstrated in Section 5, if the goal is to selectively enhance certain model capabilities—such as mathematical reasoning or coding proficiency—or to improve the model’s grasp of specific domains of knowledge, it is sufficient to apply sparse finetuning [1, 2] to only those parameters responsible for the targeted capability, while freezing the remaining paramters of the model. This approach can effectively improve task-specific performance while minimizing negative interference with unrelated capabilities. Future work should explore more precise methods for localizing knowledge within model parameters, as well as more efficient and robust techniques for sparse finetuning.
>
>
> 2. **Knowledge Clustering in the Pre-training Corpus**
>
>    During pretraining, organizing knowledge-related data fragments close together in the corpus—such as grouping Harry Potter novels, related articles, and corresponding Wikipedia entries—may promote greater parameter specialization. Based on the findings in [1, 2], we hypothesize that this approach facilitates the backpropagation of gradients associated with shared knowledge into a concentrated set of parameters, thereby enhancing the overall degree of parameter specialization.
>
>
> Our work opens up a new interpretability perspective for understanding knowledge storage in LLMs—an area that remains underexplored and calls for further investigation by the research community. In the revised version of the paper, we will include a dedicated subsection discussing these future work in more detail.
>
> ---
>
> **References**：
>
> [1] Backward Lens: Projecting Language Model Gradients into the Vocabulary Space
>
> [2] How Do Large Language Models Acquire Factual Knowledge During Pretraining?

---

> > ### Comment · Reviewer_vpb2 · 2025-08-04
> >
> > Thank the authors for the rebuttal! It addresses most of my concerns. While I do not fully agree with the authors on the first point (like studied in paper [1]), I acknowledge that zero ablation is also a widely used and acceptable approach. The optimal ablation in [1] might be too complex to implement if it is not this paper's focus, but using the average value to do ablation is worth trying. Since the rebuttal addressed most of my concerns, I would like to maintain my positive score.
> >
> > [1] Maximilian Li, Lucas Janson. Optimal ablation for interpretability. NeurIPS 2024.

---

> ### Author Response · Authors · 2025-08-06
> **Responses to Reviewer vpb2**
>
> We sincerely thank Reviewer vpb2 for the thoughtful feedback and for maintaining a positive score!
>
> We appreciate your acknowledgment of the validity of the zero-ablation method used in our work. And yes, the specific choice of ablation methods between them is not central to our paper and does not affect our main conclusions. That said, we thank you for suggesting the average-value ablation approach. We will incorporate this method into experiments in the future and also discuss it in the revised version of the paper.

---

### Decision · Program_Chairs · 2025-09-17

**Decision:**

Accept (poster)

**Comment:**

This paper introduces the interesting concept of "parameter specialization" to explain how knowledge is organized within large language models. The reviewers were in agreement about the paper's primary strengths: a large-scale, rigorous analysis across 20 open LLMs, that provides compelling evidence that model performance is highly correlated with increased parameter specialization. The work represents a significant contribution to the field of LLM interpretability by offering a new perspective and a reproducible methodology for analyzing knowledge storage patterns. The review process raised important questions about the precise interpretation of the proposed PSS metric, and the paper would benefit from additional clarifications regarding the definition of the metric and the expanded limitations section. However, the consensus is that the paper's strengths and the importance of its findings strongly warrant its acceptance.